# Composite p-Si/Al_2_O_3_/Ni Photoelectrode for Hydrogen Evolution Reaction

**DOI:** 10.3390/ma16072785

**Published:** 2023-03-30

**Authors:** Putinas Kalinauskas, Laurynas Staišiūnas, Asta Grigucevičienė, Konstantinas Leinartas, Aldis Šilėnas, Dalia Bučinskienė, Eimutis Juzeliūnas

**Affiliations:** Center for Physical Sciences and Technology, Saulėtekio av. 3, LT 10257 Vilnius, Lithuania

**Keywords:** atomic layer deposition, photoelectrochemistry, silicon, photocatalysis

## Abstract

A photoelectrode for hydrogen evolution reaction (HER) is proposed, which is based on p-type silicon (p-Si) passivated with an ultrathin (10 nm) alumina (Al_2_O_3_) layer and modified with microformations of a nickel catalyst. The Al_2_O_3_ layer was formed using atomic layer deposition (ALD), while the nickel was deposited photoelectrochemically. The alumina film improved the electronic properties of the substrate and, at the same time, protected the surface from corrosion and enabled the deposition of nickel microformations. The Ni catalyst increased the HER rate up to one order of magnitude, which was comparable with the rate measured on a hydrogen-terminated electrode. Properties of the alumina film on silicon were comprehensively studied. Grazing incidence X-ray diffraction (GI-XRD) identified the amorphous structure of the ALD oxide layer. Optical profilometry and spectroscopic ellipsometry (SE) showed stability of the film in an acid electrolyte. Resistivity measurements showed that annealing of the film increases its electric resistance by four times.

## 1. Introduction

Photoelectrochemical (PEC) hydrogen generation from water offers an opportunity for solar energy accumulation and transmission with a very low carbon footprint. There is permanent demand for efficient and low-cost photoelectrodes for hydrogen evolution reaction (HER). Silicon-based photoelectrodes are most promising due to their narrow band gap (~1.1 V), which seamlessly matches solar spectrum and efficiently absorbs photons. Silicon is a non-toxic, robust material, which is likely to dominate the photovoltaic market for the next few decades at least [1,2,3]. Advanced infrastructure is available to produce highly controllable silicon structures and compositions.

Unprotected silicon is rather active. Under ambient conditions, it forms an oxide (SiO_x_) and energetic holes on the surface. The oxide formation suppresses the current efficiency as well as increases the number of recombination pathways [4,5,6]. Oxidation can occur even during cathodic polarization due to Si interaction with dissolved oxygen [4]. To overcome such issues, an interfacial stabilization approach is needed. One such opportunity is provided by Si surface passivation by alumina (Al_2_O_3_) film. Al_2_O_3_ has shown effective Si passivation for PV applications, particularly for those which are based on p-type silicon (p-Si) [4,7,8,9,10,11,12,13,14]. These films chemically reduce the interface state density (*D*_if_); this effect is referred to as electronic passivation. The Al_2_O_3_ films typically possess a fixed negative charge, which causes the field passivation effect [12,13]. Fixed charges at the Si surface interact with the bulk charges, thereby inducing a depletion layer close to the Si surface that determines the passivation effect. Effective silicon passivation by ultrathin Al_2_O_3_ layers has been proven in solar cells modified with black silicon—a material whose light absorbance is nearly 100% [12,13].

While Si passivation by Al_2_O_3_ for PV cells has been widely studied, very little is known about the PEC functionalities of such systems in electrolytes. It is desirable that the oxide layer on Si can increase the carrier lifetime, protect the interface from corrosion, and, at the same time, sustain the rate of PEC reactions. In most cases, however, the oxide films protect silicon from corrosion, but also reduce its electrochemical activity in terms of charge transfer to electrolytes [5]. To overcome such a disadvantage, HER catalysts—predominantly platinum group metals (Pt, Ru, Rh, Ir)—are used.

Nickel has been proposed as a less costly substitute for the precious metal catalysts. Li and Wang studied the photocatalytic effect of Ni with respect to oxygen evolution on n^+^/p-Si and n^+^/n-Si electrodes in an alkaline medium [15]. A film 10–30 nm in thickness was evaporated or sputtered. The film converted into Ni(OH) upon anodic polarization. Kenney et al. reported Si photoanodes coated with a 2 nm Ni layer on an n-Si substrate for oxygen generation in alkaline media [16]. The electrode was stable within 80 h of PEC water oxidation. Feng et al. reported the deposition by e-beam evaporation of continuous layers of Ti (15 nm) and Ni (5 nm) on p-Si [17]. The authors concluded that Ni provides adequate protection of a Si photocathode in borate buffers. Protective films of Ni on Si also have certain disadvantages. A metallic film does not prevent the development of recombination centers at the surface. Additionally, metallic films do not possess the fixed charge that causes the field passivation effect. The transparency of continuous film can also be an issue which limits the efficiency of energy conversion [18].

Here, we propose a hybrid photoelectrode which is composed of Ni microcatalyst on alumina-stabilized p-Si for HER. Randomly distributed Ni microformations accelerate the HER without reducing the light-responsible area of the electrode considerably. The alumina film improves the electronic properties of the substrate and protects it from electrochemical disintegration. The electrode preparation method is based on ALD passivation of p-Si by an ultrathin alumina (Al_2_O_3_) layer and PEC deposition of Ni microformations.

## 2. Materials and Methods

P-type silicon wafers of <100> orientation with resistivity 10–30 Ω cm were used as substrates for photoelectrode preparation. Native oxide was removed from the silicon surface using H_2_O_2_ + H_2_SO_4_ (1:1) and HF (1 HF:40 H_2_O) solutions.

The Al_2_O_3_ films were deposited by ALD using a Fiji F200 reactor from Cambridge Nanotech. The deposition conditions were as follows: aluminum precursor—Trimethyl-aluminum (TMA, (CH_3_)3Al); oxygen precursor—distilled water; carrier gas—argon; temperature—200 °C; pressure in the deposition chamber—~0.33 mbar.

Grazing incidence X-ray diffraction (GI-XRD) was used to study the structural properties of the ALD film. A measurement geometry with the angle of Cu Kα beam set to 0.5° was applied. Such geometry enabled a structural study of nanofilms; it also reduced the possible influence of the substrate. The scanning step was set at 0.02°, and the scan rate was 2.5° min^−1^.

Surface topography and profiles were measured using optical profiler Contour GT-K (Bruker) in a white light interferometry (WLI) mode. Measurements were performed with 20× objective and 0.55× field of view magnification lens.

Thickness of the oxide films was evaluated using spectroscopic ellipsometry (SE). The measurements were performed using a dual rotating compensator ellipsometer in the spectral range from 300 nm to 900 nm. Thickness and surface roughness of the films were obtained by fitting the measured data using a Cody—Lorentz model.

PEC measurements were carried out in 1 M NaClO_4_ solution. The electrolyte was acidified to pH 3 using perchloric acid. Figure 1 shows the principal configuration of the PEC cell. The cell consisted of an optical (quartz) window, a Pt counter electrode, and a Hg/HgSO_4_ reference electrode. The area of the working electrode which contacted the solution was ~0.5 cm^2^. The potentials are given throughout the study with respect to NHE. LED illumination (*λ* = 505 nm, *N*~50 mW cm^−2^) was applied in the PEC experiments.

A four-point configuration was applied to measure resistance of the ALD Al_2_O_3_ films. The contacts on the samples were made by thermal evaporation using a vacuum evaporation system VAKSIS PVD-Vapour-5S Th. Rectangular contacts (1.2 mm × 3.1 mm) were formed using a template. After ALD coating, the Si-Al_2_O_3_ sample was cut into two equal parts: one for the as-deposited studies and another for the studies of annealing. By doing so, uncertainties and data scattering due to variation of substrate properties as well as the ALD procedure are minimized. In the same ALD batch, an Al_2_O_3_ layer on a glass wafer for resistance studies was also produced. The current–voltage (*I*–*U*) characteristics were measured using a U3606A multimeter/DC Power Supply from Agilent Technologies. The characteristics of two adjacent gold contacts on the Al_2_O_3_ layer were measured.

## 3. Results

The as-deposited 10 nm Al_2_O_3_ layer was amorphous; the GI-XRD data did not indicate any crystallinity features of the layer (Figure 2). Thermal treatment of the Al_2_O_3_ layer at ~400 °C normally improves its capability of electronic passivation [7,8,9,10,11,12,13,14]. It is assumed that annealing reduces the defect density *D*_it_ at the Si/Al_2_O_3_ interface. The *D*_it_ effect can be caused by several phenomena, including interfacial hydrogenation, film relaxation, rearrangements of Si-O bonds, and some additional oxide growth upon thermal treatment [11]. Annealing shifted the open circuit potential of p-Si with 10 nm Al_2_O_3_ by 0.25 V and increased the positive photoresponse at an open circuit by about 25% [19]. The GI-XRD data show that the annealing of the alumina film at 400 °C does not change its crystallographic structure; the annealed sample remains amorphous. The sample had also been immersed in the acid solution, in which the PEC study was carried out. The exposure appreciably changed the GI-XRD pattern. Thus, the Al_2_O_3_ layer retains an amorphous structure under annealing and exposure conditions.

Figure 3 compares the topographical features of the as-deposited sample (a) to those after the sample immersion in electrolyte and annealing (b). The treatment sequence for sample 4 is indicated in Table 1. The scales given in color characterize the peak-to-valley distance (∆*d*). The maximum value of the as-deposited sample is ∆*d* = 7 nm. This value also includes possible contribution from the Si substrate. The same sample after immersion and annealing is characterized by a higher value (∆*d* = 9.4 nm). The surface roughness also increases from *S*_a_ = 2.23 nm to 4.76 nm. Note that the parameter *S*_a_ (Figure 4) differs from the parameter ∆*d* (Figure 3). The parameter *S*_a_ means average roughness, which is calculated as an arithmetic average of the absolute values of the height deviations from the best-fitting plane over the complete 3D surface. The thickness of the oxide layer also increases slightly from *h* = 8.12 nm to 9.11 nm during the exposure. Altogether, these data do not indicate any evidence of the oxide dissolution. An increase in the film thickness implies an expansion of the oxide structure due to electrolyte accommodation.

**Table 1 materials-16-02785-t001:** Topographical data obtained for the p-Si ALD coated with 10 nm Al_2_O_3_ layer and after sample exposure to 1 M NaClO_4_ (pH 3) and annealing at 400 °C. The film thickness (*h*) and roughness (*S*_a_) were determined by SE; meanings of these parameters are explained in Figure 4. The values are given in nanometers. Average values from four measurement series (nos. 1–4) are presented.

Sample No.	1	2	3	4
Procedures	p-Si ALD coated with 10 nm Al_2_O_3_	Sample 1 after immersion in 1 M NaClO_4_ (pH 3) for 30 min	Sample 2 after annealing at 400 °C for 30 min	Sample 3 after immersion in 1 M NaClO_4_ (pH 3) for 30 min
Al_2_O_3_ film thickness (*h*)	8.12	9.11	9.81	9.66
Roughness (*S*_a_)	2.23	3.83	3.26	4.76

Electrical resistance (*R*) of a passive film is an important factor which determines the rate of electron transfer from electrode to electrolyte (and vice versa), thereby affecting the corrosion rate. We studied the resistance of alumina layer using the measurement whose configuration is given in Figure 5. The *R*-values were derived from the slope of the current–voltage (*I*–*U*) curve close to zero current (*R*_1_ = (Δ*U*/Δ*I*)Δ*U* → 0) where the curves are rectilinear. At higher voltages, the characteristics deviate from linearity. Such deviation is due to the potential barriers at the junctions Au–Al_2_O_3_–Si–Al_2_O_3_–Au. The resistance across the Al_2_O_3_ film (*R*_1_) was of the order of kΩ. The resistance along the film (*R*_2_) measured on glass exceeded 6 MΩ. The resistance of the p-Si substrate between the contacts (*R*_3_) was negligible, as a rather conductive wafer (10–30 Ω cm) had been used. The resistance of the Al_2_O_3_ film between two adjacent contacts was determined. Average resistance value of the as-deposited Al_2_O_3_ film was *R*_1_ = 2.41 kΩ, whereas the value for the film annealed at 400 °C was *R*_1_ = 9.57 kΩ. This suggests that, due to a water precursor used in ALD, hydrogen-related channels are present in the as-deposited Al_2_O_3_ layer, which partially shunt either the layer itself or potential junction barriers. Annealing of the alumina film leads to suppression of the electron transfer rate.

The PEC studies (Figure 6) had been carried out using *λ* = 505 nm illumination. The applied photon energy (2.45 eV) significantly exceeded the band gap of silicon (*E*_g_ = 1.1 eV). The band gap of the dry ALD Al_2_O_3_ is even higher (*E*_g_ = 6.2 eV [20]); thus, the oxide could retain its high insulating capability in the electrolyte. Note also that the 10 nm Al_2_O_3_ layer is almost completely transparent under the applied illumination conditions. The transparency of the layer was evaluated as high as *T*r = 0.976 using a reflectance calculator that takes into account Fresnel equations (www.filmetrics.com/reflectance-calculator, accessed on 25 January 2023).

Figure 6a shows the HER photocurrents determined for the as-deposited and annealed electrodes. On both electrodes, the reaction is rather sluggish in a wide range of polarizations; remarkable photocurrents appear at *E* < −0.5 V, that is, at much more negative potentials than thermodynamics predicts. The photo-generated currents are quite low, well below one milliamp. The inset in Figure 6a shows the photocurrents measured for hydrogen-terminated silicon (Si-H) which has been prepared by removing the native oxide in HF solutions. The photocurrents on this electrode are more than one order of magnitude higher when compared to the photocurrents on the Al_2_O_3_-passivated electrode. Additionally, the photocurrent on the Si-H electrode starts at polarization *E* ~ −0.1 V, which is in accordance with the thermodynamic condition (according to Nernst equation, reaction 2H^+^ + 2e^−^ → H_2_ is characterized by *E*^0^ = −0.118 V at T = 25 °C and pH 3, or 59 mV change per pH unit). These data confirm high passivity of the p-Si photoelectrode coated with Al_2_O_3_.

It is also evident from Figure 6a that annealing increases the electrode passivity. The photocurrents on the thermally treated electrode are several times lower when compared to those measured on the as-deposited electrode. The main reason for the thermal effect lies in an increase in the film resistance, which creates a higher barrier for electron transfer to the electrolyte. As discussed above, annealing increased the oxide resistance by about four times. Such a ratio is comparable with the ratio of the corresponding photocurrents (Figure 6a).

The ultrathin alumina film provides an opportunity for localized PEC deposition of a metal catalyst. Typically, the passive films on crystalline substrates are not completely uniform and homogeneous. In electrolyte, local corrosion sites can, first of all, occur at the “weak” sites, such as structural defects, micro-cracks, etc. The surface inhomogeneities can play the role of a certain matrix for the localized deposition of a metal catalyst. Figure 6b shows the voltammetric diagram, which characterizes the PEC deposition of Ni. The photocurrents also include HER contributions. Obviously, extensive photocurrents are induced when Ni^2+^ is added to the electrolyte. The currents drop to negligible values when illumination is terminated. The photocurrents in the electrolyte without Ni^2+^ are negligible when compared to those in Ni^2+^-containing electrolyte.

Figure 7 shows the surface features of the Si-Al_2_O_3_ electrode polarized in the Ni^2+^-containing electrolyte. The Ni deposits are randomly distributed over the electrode surface; the distance between the formations is in the order of tens of micrometers. Image (b) in the figure takes a closer look at a single Ni formation. It has the shape of a microcone surrounded by a corrosion pit. Pitting formation is hardly possible under the applied polarization, which protects the substrate cathodically. The sites should develop after the termination of cathodic current. This can occur in the electrolyte or outside when drying the sample in atmosphere. By electrochemical nature, such a process is similar to the process known as a metal-assisted chemical etching (MACE) of silicon.

The formations deposited in Ni^2+^-containing electrolytes have also been studied by SEM–EDS. Figure 8 provides the analytical information; Ni microformations randomly distributed over the Si-Al_2_O_3_ surface are clearly identified.

Figure 6c demonstrates the catalytic effect of Ni microformations on PEC hydrogen reduction. The thermally treated Si-Al_2_O_3_ sample was Ni-modified by the PEC deposition, as shown in Figure 7, and then the sample was transferred to a Ni^2+^-free bath. The catalyst increases the photocurrents of hydrogen reduction up to one order of magnitude. The catalytic photocurrents are comparable with those generated on the bare, that is, the hydrogen-terminated electrode (Figure 6a, inset). Such an unprotected electrode, however, reduces its activity in the course of exposure and corrosion due to the formation of a highly insulating SiO_2_ layer [5].

## 4. Discussion

The applied method is schematically demonstrated in Figure 9. The amorphous 10 nm Al_2_O_3_ layer formed by ALD on p-Si wafer showed high stability in acid perchlorate electrolyte. The layer thickness did not show any evidence of dissolution; on the contrary, spectroscopic ellipsometry identified an increase in the layer thickness by about 10% due to electrolyte intake. The following effects have been observed from the application of an Al_2_O_3_ layer: (i) chemical passivation of the silicon surface, thereby reducing the interface state density (*D*_if_) and improving its electronic properties; (ii) protection of the silicon from interaction with electrolyte and spontaneous formation of isolating silica layer; (iii) provision of a template for photoelectrochemical deposition of Ni microcatalyst, as Figure 9a demonstrates. The proposed approach yielded randomly distributed microcatalyst over the electrode surface.

Figure 9b explains the catalytic effect of Ni with respect to the HER. The sample produced in (a) has been transferred to a Ni^2+^-free electrolyte. When illuminated, the electron supply and the hydrogen evolution are concentrated at the sites of the Ni microcatalyst. The catalyst increased the HER rate by approximately one order of magnitude, which is comparable with the reaction rate on a coating-free (hydrogen-terminated) electrode.

From a technological perspective, the results suggest that the catalytic effect can be achieved simply by introducing Ni^2+^ ions into the electrolyte in which hydrogen is produced. By selecting a Ni^2+^ concentration, a microcatalytic structure can be photoelectrochemically formed within the Al_2_O_3_ matrix. The electrolyte can be continuously replenished with the Ni^2+^ ions if necessary.

Our study has been performed in an acid electrolyte (pH 3) in order to demonstrate the principal opportunity of the acceleration of free proton reduction (2H^+^ → H_2_). It is worth mentioning that nickel and its oxides are not stable in an acid environment. However, this aspect can be neglected at high cathodic polarizations, such as those applied in our experiments. At higher cathodic polarizations, Ni is deposited and remains stable in a metallic state. Other non-precious catalysts (e.g., Mo) can also be deposited according to the proposed approach.

A variety of surface architectures can be obtained by the proposed methodology when tuning relevant factors, such as illumination duration and intensity, electrode polarization, electrolyte composition, etc. The proposed methodology is technically simple and practical. Further research is needed to elucidate the long-term behavior of the proposed photoelectrode in order to reach the technological maturity of the process.

## 5. Patents

Patent is pending; application number LT2023 503; 23 January 2023.

## Figures and Tables

**Figure 1 materials-16-02785-f001:**
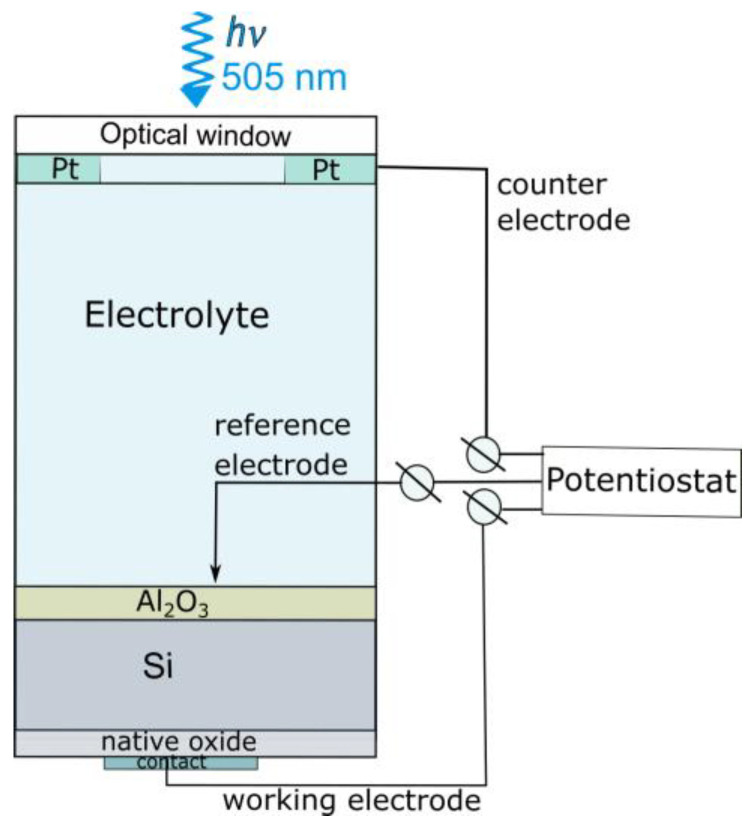
Configuration of the PEC cell consisting of the optical window, the electrolyte, the photoelectrode, and the counter and reference electrodes.

**Figure 2 materials-16-02785-f002:**
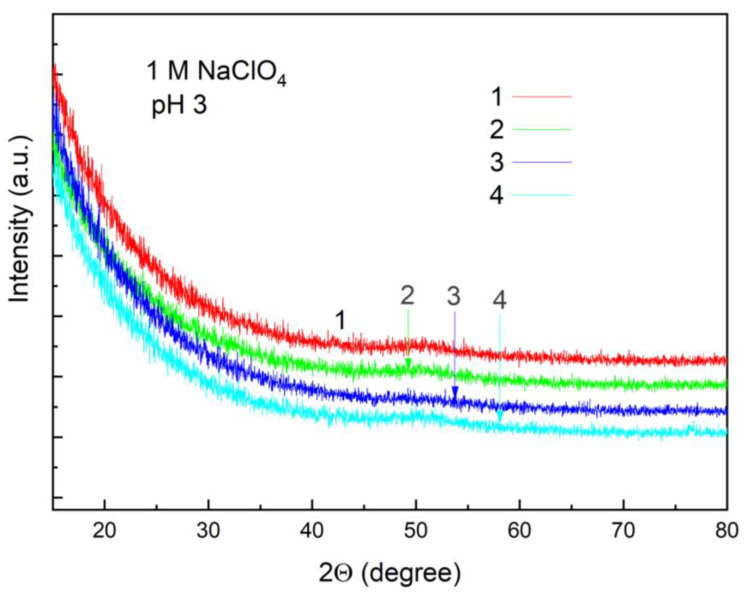
GI-XRD data for the p-Si ALD coated with 10 nm Al_2_O_3_ (1) and after immersion for 30 min in 1 M NaClO_4_ (pH 3) (2), drying and annealing at 400 °C (3), and repeated exposure to the electrolyte (4). The curve numbers correspond to the numbering in Table 1.

**Figure 3 materials-16-02785-f003:**
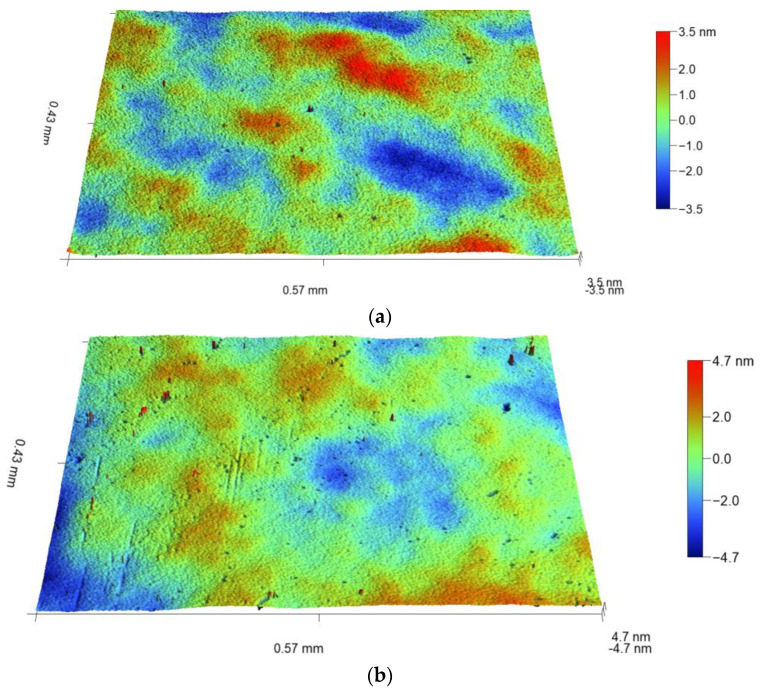
(**a**) Topography of the Si coated with 10 nm Al_2_O_3_ layer (sample No. 1, Table 1); (**b**) the sample after immersion in electrolyte and annealing (sample No. 4, Table 1). The scales in color characterize the peak-to-valley distance (∆*d*).

**Figure 4 materials-16-02785-f004:**
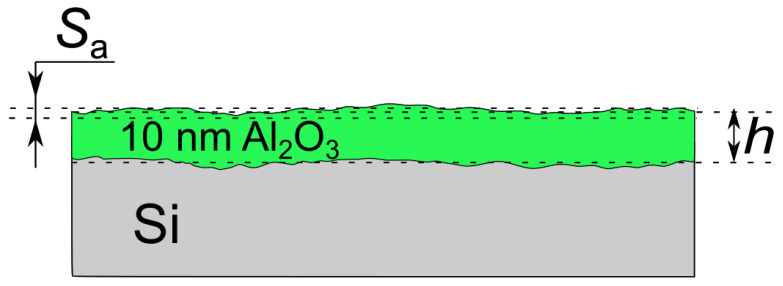
Schematic representation of Al_2_O_3_ layer on Si: S_a_ means average roughness, which is calculated as an average of the height deviations from the best fitting plane over the surface; *h* stands for an average thickness of the layer.

**Figure 5 materials-16-02785-f005:**
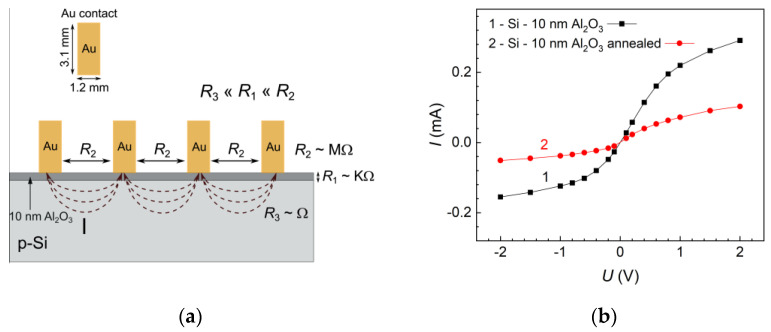
Configuration of the resistance (*R*) measurement of 10 nm Al_2_O_3_ film on p-Si. Four rectangular Au contacts (1.2 mm × 3.1 mm) at 1.9 mm distance from each other had been prepared by thermal evaporation using a template (**a**). *I*–*U* curves were measured between two adjacent contacts for the as-deposited (1) and the annealed (2) samples (**b**). The resistances were derived from the slope of the curve around zero current.

**Figure 6 materials-16-02785-f006:**
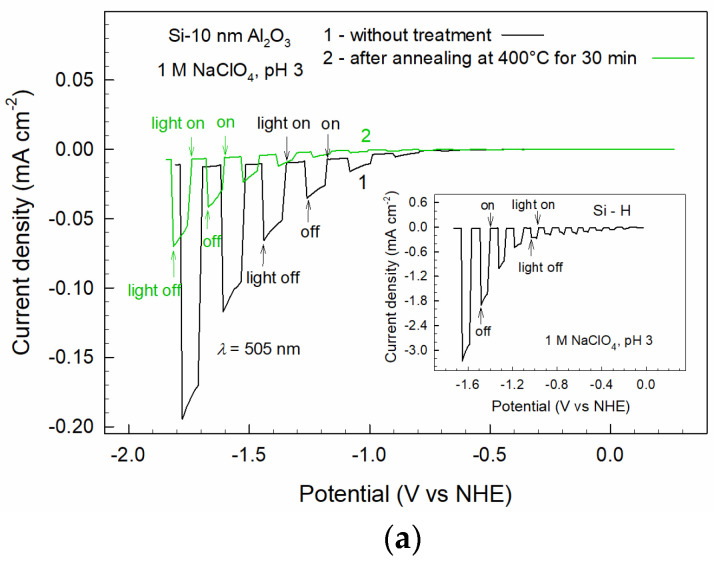
(**a**) Cathodic photocurrents of HER in 1 M NaClO_4_ (pH 3) measured for p-Si with 10 nm Al_2_O_3_ layer (1) and that annealed for 30 min at 400 °C (2); the inset shows the photocurrents of the oxide-free hydrogen-terminated silicon (p-Si-H) obtained after removing the native oxide in H_2_O_2_ + H_2_SO_4_ (1:1) and HF (1 HF:40 H_2_O) solutions; (**b**) the PEC curve (2) obtained when added into electrolyte 0.1 M NiSO_4_ (pH 3); (**c**) the PEC measurement for the p-Si-Al_2_O_3_ electrode with the Ni microformations (Figure 7). The illumination (*λ* = 505 nm, *N* = 50 mW cm^−2^) was switched on and off during the potential scan at the rate of 5 mV s^−1^.

**Figure 7 materials-16-02785-f007:**
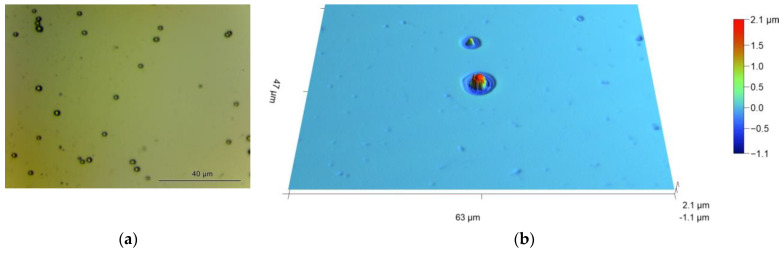
Surface views obtained by optical profilometry for Si with 10 nm Al_2_O_3_ layer after PEC Ni deposition in 1 M NaClO_4_ + 0.1 M NiSO_4_ (pH 3): (**a**) randomly distributed Ni deposits (dark spots); (**b**) single microcones surrounded by corrosion pits. Ni was deposited under potentiodynamic conditions as shown in Figure 6b.

**Figure 8 materials-16-02785-f008:**
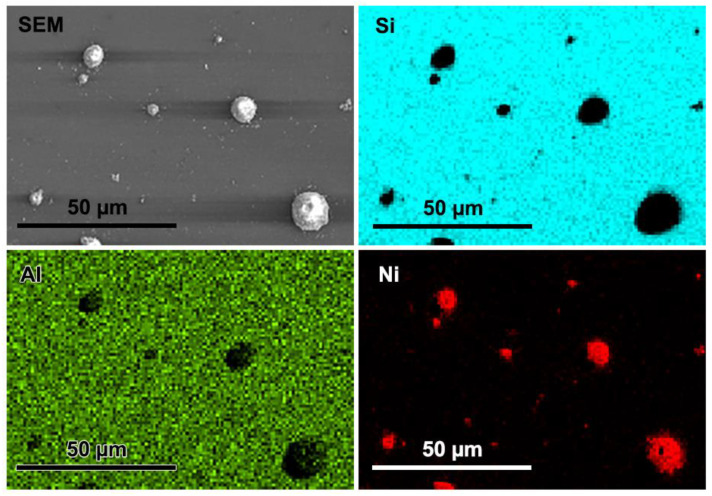
SEM–EDS images of the p-Si coated with 10 nm Al_2_O_3_ layer after photoelectrochemical Ni deposition in 1 M NaClO_4_ + 0.1 M NiSO_4_ electrolyte (pH 3). Images identify randomly distributed Ni microformations. The elements are identified by color: cyan—silicon, green—aluminum, red—nickel. Ni was deposited under potentiodynamic conditions, as outlined in Figure 6b.

**Figure 9 materials-16-02785-f009:**
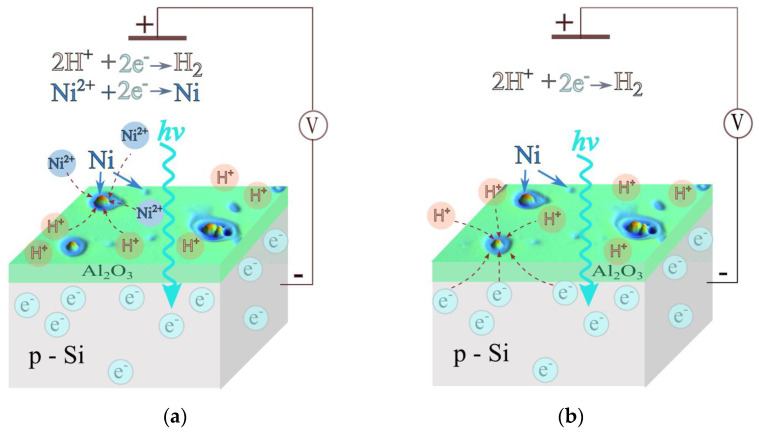
Schematic representation of Ni^2+^ deposition (**a**) and H^+^ reduction (**b**) on Ni microcatalyst on illuminated p-Si-Al_2_O_3_ electrode.

## Data Availability

The data presented in this study are available on request from the corresponding author.

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
