# Peer review of "Composite p-Si/Al2O3/Ni Photoelectrode for Hydrogen Evolution Reaction"

_materials, 2023, doi:10.3390/ma16072785_

Round 1

Reviewer 1 Report

In this paper, a photoelectrode for hydrogen evolution reaction (HER) is proposed, which is based on p-type silicon (p-Si) passivated with ultrathin (10 nm) alumina (Al2O3) layer and modified with microformations of nickel catalyst. The Al2O3 layer was formed by the atomic layer deposition (ALD), while nickel was deposited photoelectrochemically. The alumina film improved the electronic properties of the substrate and, at the same time, protected the surface from corrosion and enabled the deposition of nickel microformations. The Ni catalyst increased the HER rate up to one order of magnitude, which was comparable with the rate measured on a hydrogen-terminated electrode. Properties of the alumina film on silicon have comprehensively been studied. Grazing incidence X-ray diffraction (GIXRD) identified the amorphous structure of the film. Optical profilometry and spectroscopic ellipsometry (SE) showed stability of the film in an acid electrolyte. Resistivity measurements have shown that annealing of the film increases its electric resistance by four times.

However, there are some questions and suggestions. I suggest it can be considered after revisions.

 (1)   Some tests, such as TEM, SEM of the photoelectrode should be tested and discussed.

(2)   The stability of the photoelectrode should be discussed.

(3)   It should be given a comparisons with other photoelectrodes based on Si.

Author Response

We are grateful for the reviewer’s valuable comments.

Reviewer: (1) Some tests, such as TEM, SEM of the photoelectrode should be tested and discussed.

Answer: SEM–EDX images have been added, which provide analytical data on the surface structure (Figure 8).

Reviewer: (2) The stability of the photoelectrode should be discussed.
Answer: We have stated in the final part that further research is needed to elucidate the long-term behavior of the proposed photoelectrode in order to reach the technological maturity of the process. Unfortunately, at present we cannot offer such study. A new PEC flow cell has to be designed and manufactured, in order not to change the electrolyte pH during the prolonged experiment. Of course, we will include such study in our future plans.
Reviewer: (3) It should be given a comparisons with other photoelectrodes based on Si.

Answer: As the cited review paper (Ref. [2]) shows, there are many Si-based photoelectrodes studied under various experimental conditions. We have compared under identical conditions the activity of the Si/Al2O3/Ni electrode with the activity of a coating-free electrode (Figure 6(a)). A conclusion was drawn that Ni microcatalyst increased the HER rate up to one order of magnitude, and such rate was comparable with that measured on a coating-free (hydrogen-terminated) electrode.

Reviewer 2 Report

The manuscript "Composite p-Si/Al2O3/Ni photoelectrode for hydrogen evolu-2 reaction" is very interesting and can be accepted after addressing the following comments.

1. The authors should change x axis Intensity (a.u.) and 2ϴ (degree), XRD can be analyzed at low scan rate peaks and will get well sharp peaks.

2. Figure 3 should insert high-quality images in PNG/Tiff format.

3.  The authors should cite the following articles at appropriate places, https://doi.org/10.1016/j.ijhydene.2021.11.171, https://doi.org/10.1016/j.chemosphere.2022.137030, https://doi.org/10.1016/j.ijhydene.2022.07.115,

4. The authors should check the stability of the optimized photoelectrode at identical conditions
.

5. The authors, merge figures 6,7,9, and stability for a better understanding to readers.  

6. The Authors, should be inserted in abstracts, with Remarkable results values.

7. Figure 10 (a-b) is a little bit confusing for readers, it clearly, represents a Plausible reaction mechanism.

8.   The authors should carefully check grammatical sentences in some places are inappropriate and reference suffixes and prefixes and the authors should follow the template of the materials journal. 

Author Response

We are very grateful for the valuable comments.

Reviewer: 1. The authors should change x axis Intensity (a.u.) and 2ϴ (degree), XRD can be analyzed at low scan rate peaks and will get well sharp peaks.

Answer: We corrected the axis. The XRD measurements have been performed at the rate of 2.5° min-1. This rate is more than sufficient to identify the possible peaks. Note also that sharpness of the peaks does not depend on the scan rate; they depend predominately on the crystallite size.

Reviewer: 2. Figure 3 should insert high-quality images in PNG/Tiff format.

Answer: The image have been prepared in a Tiff format and inserted into template.

Reviewer: 3.  The authors should cite the following articles at appropriate places, https://doi.org/10.1016/j.ijhydene.2021.11.171, https://doi.org/10.1016/j.chemosphere.2022.137030, https://doi.org/10.1016/j.ijhydene.2022.07.115,

Answer: The Editor has recommended that these references are not mandatory and we can choose if we would like to refer to them or not. We chose the second option. Sorry for that.

Reviewer: 4. The authors should check the stability of the optimized photoelectrode at identical conditions.

Answer: It was given at the end part of the manuscript: “further research is needed to elucidate the long-term behavior of the proposed photoelectrode in order to reach the technological maturity of the process”. This is, however, a time-consuming task, we are not able to complete it during reasonable time-span to meet the 10 days requirement for the revision. We included such experiment in our future plans.

Reviewer: 5. The authors, merge figures 6,7,9, and stability for a better understanding to readers.  

Answer: We merged the figures and made necessary changes in the text.

Reviewer: 6. The Authors, should be inserted in abstracts, with Remarkable results values.

Answer: It is hard to understand this requirement. If it comes to quantitative values, some of them are clearly stated in the abstract: i) The Ni catalyst increased the HER rate up to one order of magnitude; ii) resistivity measurements have shown that annealing of the film increases its electric resistance by four times.

Reviewer: 7. Figure 10 (a-b) is a little bit confusing for readers, it clearly, represents a Plausible reaction mechanism.

Answer: It is difficult to understand this comment.

Reviewer: 8. The authors should carefully check grammatical sentences in some places are inappropriate and reference suffixes and prefixes and the authors should follow the template of the materials journal. 

Answer: A native English speaker has checked the text. We would be very grateful if the reviewer could identify inappropriate sentences. We improved the list of references in accordance with the requirements of this journal.  

Round 2

Reviewer 1 Report

It can be accepted.

Reviewer 2 Report

na